# Updated Clinical Evaluation of the CLUNGENE^®^ Rapid COVID-19 Antibody Test

**DOI:** 10.3390/healthcare9091124

**Published:** 2021-08-30

**Authors:** Christopher C. Lamb, Fadi Haddad, Christopher Owens, Alfredo Lopez-Yunez, Marion Carroll, Jordan Moncrieffe

**Affiliations:** 1Weatherhead School of Management, Case Western Reserve University, 10900 Euclid Ave., Cleveland, OH 44106, USA; 2Silberman College of Business, Fairleigh Dickinson University, 1000 River Rd., Teaneck, NJ 07666, USA; 3BioSolutions Services LLC, 92 Irving Avenue, Englewood Cliffs, NJ 07632, USA; 4Fellow of the Infectious Disease Society of America (IDSA), 4040 Wilson Boulevard, Suite 300, Arlington, VA 22203, USA; 5Fadi Haddad, MD, Inc., 8860 Center Dr. Suite 320, La Mesa, CA 91942, USA; Fadi.Haddad@sharp.com; 6Sharp Grossmont Hospital, 5555 Grossmont Center Dr., La Mesa, CA 91942, USA; 7Alivio Medical Center, Indianapolis, IN 46219, USA; aowens@aliviomedicalcenter.com (C.O.); alopezyunez@aliviomedicalcenter.org (A.L.-Y.); 8MedComp Sciences, LLC, 20503 MacHost Road, Zachary, LA 70791, USA; Marion.Carroll@medcompsciencesllc.com (M.C.); Jordan.Moncrieffe@medcompsciencesllc.com (J.M.)

**Keywords:** COVID-19 antibody testing, COVID-19 immunity, COVID-19 serology, CLUNGENE^®^

## Abstract

Background: COVID-19 antibody testing has been shown to be predictive of prior COVID-19 infection and an effective testing tool. The CLUNGENE^®^ SARS-COV-2 VIRUS (COVID-19) IgG/IgM Rapid Test Cassette was evaluated for its utility to aide healthcare professionals. Method: Two studies were performed by using the CLUNGENE^®^ Rapid Test. (1) An expanded Point-of-Care (POC) study at two clinical sites was conducted to evaluate 99 clinical subjects: 62 positive subjects and 37 negative subjects were compared to RT-PCR, PPA, and NPA (95% CI). Sensitivity was calculated from blood-collection time following symptom onset. (2) A cross-reactivity study was performed to determine the potential for false-positive results from other common infections. Results: The specificity of subjects with confirmed negative COVID-19 by RT-PCR was 100% (95% CI, 88.4–100.0%). The sensitivity of subjects with confirmed positive COVID-19 by RT-PCR was 96.77% (95% CI, 88.98–99.11%). In the cross-reactivity study, there were no false-positive results due to past infections or vaccinations unrelated to the SARS-CoV-2 virus. Conclusion: There is a need for a rapid, user-friendly, and inexpensive on-site monitoring system for diagnosis. The CLUNGENE^®^ Rapid Test is a useful diagnostic test that provides results within 15 min, without high-complexity laboratory instrumentation.

## 1. Introduction

The COVID-19 pandemic SARS-CoV2 COVID-19 has infected over 140 million people worldwide and has caused approximately 3.89 million deaths as of 23 June 2021 [1]; however, some studies suggest that the actual number of global COVID-19 deaths may be about 6.9 million, which is more than double the recorded amount [2,3,4]. In response to the pandemic, the US Food and Drug Administration (FDA) authorized the use of COVID-19 serological tests through Emergency Use Authorizations (EUAs) to make COVID-19 in vitro diagnostic tests widely available to help identify individuals with an adaptive immune response indicating recent or prior infection [5]. Serology tests, or immunoassays, play a significant role in the fight against COVID-19 [6,7]. A prior history of SARS-CoV-2 infection is associated with a lower risk of infection, with an estimated protective effect of up to seven (7) months following primary infection; this supports the conclusion that convalescent plasma with specific antibodies to SARS-CoV-2 has powerful antiviral activity, which can reduce the viral load and mortality in patients with active COVID-19 infection [8,9,10].

There is an urgent need for a rapid, user-friendly, and inexpensive on-site monitoring system for diagnosis [11]. The CLUNGENE^®^ SARS-COV-2 VIRUS (COVID-19) IgG/IgM Rapid (15 min) Test Cassette has been commercially available in the US under an FDA-approved Emergency Use Authorization (EUA201121) [12] and Europe (CE Mark reference 02PBJ267 dated 9 March 2020). The CLUNGENE^®^ test has been previously studied, including the use of the test in the offices of general practitioners, evaluating the presence of antibodies in convalescent plasma donors, and how the test performs at a point of care facility [13,14,15]. The aim of this research was to better understand the sensitivity and specificity of the CLUNGENE^®^ assay and the potential for false-positive results related to infections or vaccinations not linked to the SARS-CoV-2 virus.

## 2. Materials and Methods

### 2.1. Study #1

#### 2.1.1. Design

In an initial study, a single Point-of-Care (POC) facility was used to estimate the sensitivity and specificity of the CLUNGENE^®^ test [16]. The study was expanded to a second independent site. The two sites were Sharp Healthcare, a not-for-profit multi-center regional healthcare group located in San Diego, CA; and Alivio Medical Center, an urgent care/primary care center in Indianapolis, IN. Samples used for RT-PCR were nares swabs. Positive subjects were symptomatic for SAR-CoV-2 Virus infected and confirmed with RT-PCR positive tested nares swabs. Negative subjects were asymptomatic, from high-risk areas, and confirmed with negative RT-PCR tested nares swabs. Finger-prick whole-blood samples were used for SARS-COV-2 Virus IgG/IgM detection. The comparator method for RT-PCR was either Cobas Roche SARS-COV2 RT-PCR (Roche Diagnostics, 9115 Hague Road PO Box 50457, Indianapolis, IN 46250) or Thermo Fisher TaqPath COVID-19 Combo Kit (Thermo Fisher Scientific, 168 3rd Ave, Waltham, MA 02451, USA). Trained operators with no prior information about each subject drew samples. Subject inclusion criteria included individuals with a confirmed COVID-19 test result by SARS-CoV-2 RT-PCR. Subjects were excluded if they were unable to provide informed consent due to mental or cognitive disabilities.

#### 2.1.2. Methods

The CLUNGENE^®^ Point-of-Care test was run according to the manufacturer’s instructions (see Figure 1). The test result was read after 15 min. Days from symptom onset were captured from an electronic medical record which documented self-reported data from patients reporting on the number of days they had been sick at the time of study enrollment.

Categorical variables were compared using the chi-squared or Fisher exact test, and continuous variables were compared using the Student *t*-test or Mann-Whitney U test, as appropriate. All tests were two tailed, and *p* < 0.05 was considered statistically significant. SPSS Statistics, IBM SPSS software, version 27.0 (SPSS, Inc., Chicago, IL, USA) was used for all calculations.

The CLUNGENE^®^ test can be stored at ambient temperature between 4 and 30 degrees Celsius. The product is packaged in a box with 25 test kits, with each kit wrapped in an individual foil patch for proper storage and transportation. The test kit itself should be kept away from direct sunlight.

### 2.2. Study #2

#### 2.2.1. Design

An initial cross-reactivity study was performed in which various common infectious agents were tested for potential false-positive results (Table 1). All tests were negative. In addition, as recommended by the US Food and Drug Administration [17], a follow-up study tested the cross-reactivity of the CLUNGENE^®^ device to antibodies to common coronaviruses that are not SARS-CoV-2: anti-229E, anti-NL63, anti-OC43, and anti-HKU1; and those for which there is a high rate of vaccinations and/or infection in the US, i.e., anti-Haemophilus influenzae IgG and IgM. The testing was performed at Medcomp Sciences, an independent clinical medical laboratory [18].

#### 2.2.2. Methods

The CLUNGENE^®^ Point-of-Care test was run according to same manufacturer’s instructions as was performed with Study #1. Samples (50) used for the cross-reactivity study were obtained from Trina Bioreactives, Ag; Trina Bioreactives, Ag, (Grabenstrasse 8, 8606 Nänikon, Switzerland) is an ISO certified company providing specialized Sars-COV-2 in vitro diagnostic biomaterials [19]. The samples were collected under a protocol approved by the ethics committee of the National Medical Association, Baden-Wurttemburg, Germany (file #F-2012-027, “Plasma Samples for Studies”).

Test results were read after 15 min. Five (5) serum samples that were positive for IgM and five for IgG were analyzed in three (3) separate CLUNGENE^®^ Point-of-Care test lots with each of the below antibodies:Anti-Haemophilus influenzae IgM,Anti-Haemophilus influenzae IgG,IgG, anti-coronavirus 229E IgG,Anti-coronavirus NL63 IgG,Anti-coronavirus OC43 IgG,Anti-coronavirus HKU1 IgG,Anti-coronavirus 229E IgM,Anti-coronavirus NL63 IgM,Anti-coronavirus OC43 IgM,Anti-coronavirus HKU1 IgM.

Cross-reactivity was determined by using Beckman Coulter UniCel DxI Access Immunoassay System is an in vitro diagnostic device used for the quantitative, semi-quantitative, or qualitative determination of various analyte concentrations found in human body fluids (Table 2).

All QC materials were supplied and used as indicated by the manufacturer. A minimum of 2 levels of QC were processed per run in accordance with Westgard rules for 6-sigma quality requirements. QC results must fall within 3 standard deviations of historical data, as recorded in Levy–Jennings plots by the system. All QC results were documented and verified by Clinical Laboratory Scientist before processing samples (Table 3).

Qualitative results are reported as Positive/Reactive or Negative/Non-Reactive, depending on whether the analyte in question is above or below the established signal cutoff value (S/CO). Each analyte tested may have a different cutoff value, but each cutoff value is determined during calibration of the instrument, using the value of the calibrator.

## 3. Results

### 3.1. Results of Study #1

An analysis was run on 99 patients who completed the study (Table 4).

Thirty-seven (37) patients who had negative COVID-19 RT-PCR were tested and found antibody negative by using the CLUNGENE^®^ test (95% CI, 90.60~100.00%). Three (3) out of four (4), or 75%, of RT-PCR positive COVID-19 subjects tested prior to day 7 of symptom onset were antibody positive (95% CI, 30.06~95.44%). Twenty-three (23), or 100%, of RT-PCR positive COVID-19 subjects tested positive between day 8 and 14 from symptom onset (95% CI, 85.69~100.00%). A total of 33 out of 35, or 94.28%, of RT-PCR positive COVID-19 subjects tested positive after day 14 from symptom onset were antibody positive (95% CI, 30.06~95.44%). In all 62 patients with confirmed COVID-19 with RT-PCR, the combined sensitivity of IgM and IgG was 96.77% (95% CI, 88.98–99.11%), meaning that there was 96.77 positive agreement between a positive RT-PCR test and a positive antibody test. The specificity was 100% (95% CI, 88.4–100.0%), meaning there was 100% agreement between a negative RT-PCR test and 100% negative antibody result. These results are displayed in Table 5.

The positive predictive value can be calculated, but the result is dependent on the prevalence of disease in the community. If a test for a disease has 96.77% sensitivity and 100% specificity, and the disease prevalence is 10%, the positive predictive value (PPV) is 100%, and the negative predictive value (NPV) is 99.64%. At the time of this publication, the positivity rate for the nares SARS CoV2 RT-PCR tests was between 8.2% and 10% in San Diego, CA (San Diego County), and Indianapolis, IN (Marion Country) [20].

### 3.2. Results of Study #2

The test results of negative quality-control samples were all negative, and the test results of the positive quality-control samples were all positive. The consistency rate of cross-reactivity of negative samples was 100%.

## 4. Discussion

In the first study, the performance characteristics of CLUNGENE^®^ were evaluated and showed a specificity of 100% and a sensitivity of 96.77%. In the second study, there were no false-positive results due to past infections or vaccinations unrelated to the SARS-CoV-2 virus. These results are in line with the new European Commission’s Medical Device Coordination Group requirements for rapid COVID-19 antibody tests and consistent with previously published results [21,22,23,24,25].

Antibody testing is a useful aid to confirm past infection [26]. Recent findings confirm that antibody testing is predictive of prior COVID-19 infection, and rapid screening methods—even from finger pricks—are effective testing tools [27]. However, we see the potential for a much broader use and recommend a combined approach that uses both RT-PCR and serological testing. The advantage of the CLUNGENE^®^ antibody test is its simplicity, since there is no need for specialized laboratory personnel to perform and interpret results. The low rate of false positivity makes this test ideal to rule in disease and eliminate the need for further RT-PCR testing if seroconversion occurs, since the CLUNGENE^®^ antibody test can diagnose most infected COVID-19 patients. If the test is negative, a recommendation should be made to have a follow-up RT-PCR test.

Serology testing also has the potential to monitor the presence of antibodies. Studies confirm that a prior history of SARS-CoV-2 infection is associated with a lower risk of infection, with an estimated seven (7) month protective effect [9,10]. The association of SARS-CoV-2 Seropositive Antibody Test with Risk of Future Infection has now been established [28]. It is clear that titers of IgM and IgG antibodies against the receptor-binding domain (RBD) of the spike protein of SARS-CoV-2 decrease significantly between one (1) and seven (7) months, and concurrently, neutralizing activity decreases [29]. Monitoring the presence of antibodies from past infection could assist healthcare professionals in assessing the likely presence of neutralizing activity and immunity when managing patient care.

Given widespread availability, COVID-19 serology testing, similar to other infectious diseases, can now become routine [30]. Consideration should be given to adding COVID-19 antibody testing to the WHO List of Essential In Vitro Diagnostics (EDL), which now consists of 122 test categories, including most serious infectious diseases [31]. A rapid 15-min COVID-19 assay offers the ability to do this testing quickly and efficiently—tests are now available for less than $5 and routinely reimbursed in the US by public and private insurers [5].

In addition, the ability of the CLUNGENE^®^ antibody test to detect antibodies to the coronavirus’s spike protein’s receptor binding domain means it has the potential to assess the efficacy of most vaccines, as well as convalescent plasma therapy [32]. Countries in Europe are now using antibody testing to determine if a second COVID-19 vaccine dose is required if a patient has a prior infection based on a positive antibody test [33]. Recently, airports and Blood Banks have been providing COVID-19 antibody testing services to determine whether a person has developed immunity to COVID-19 through vaccination or through contracting the virus previously [34,35]. Some countries, including China, require an antibody test. Limited evaluation of the CLUNGENE^®^ antibody test has confirmed positive antibody test results following patients who have been vaccinated [36]. Pfizer’s recent data suggest that its vaccine is efficacious for only 6 months and that a third shot within 12 months is likely needed [37]. Furthermore, there is a potential issue regarding vaccine efficacy for recipients who do not receive a full dose; the US CDC estimates that 3% of vaccinated people who received a first dose did not receive a recommended second dose [38]. COVID-19 vaccination also fails to stimulate an immune response in many blood-cancer patients or those otherwise immunocompromised [39,40]. Additional studies are needed to confirm the efficacy of serology testing to monitor vaccine effectiveness.

### Limitations

Limitations of the study include a small sample size from two geographic areas. The study also did not include special groups, such as pregnant women or children. The subjectivity of symptom reporting by patients can be a confounding factor in determining the duration of illness. Some patients may have been symptomatic for a different time period than they recalled. Lastly, the CLUNGENE^®^ antibody test has not been compared with another test in the study.

## 5. Conclusions

In a pandemic crisis with significant economic and health implications, this study confirms the utility of serological testing for COVID-19 disease diagnosis providing rapid test results with a relatively high degree of sensitivity and specificity. Furthermore, given recent data regarding the relationship between positive serology and immunity, routine testing can be a useful tool to monitor antibody status for optimal patient care. Tests such as the CLUNGENE^®^ SARS-COV-2 VIRUS (COVID-19) IgG/IgM Rapid Test Cassette can assist healthcare professionals to help identify individuals with an adaptive immune response indicating recent or prior infection, as intended by the US FDA under an EUA.

## Figures and Tables

**Figure 1 healthcare-09-01124-f001:**
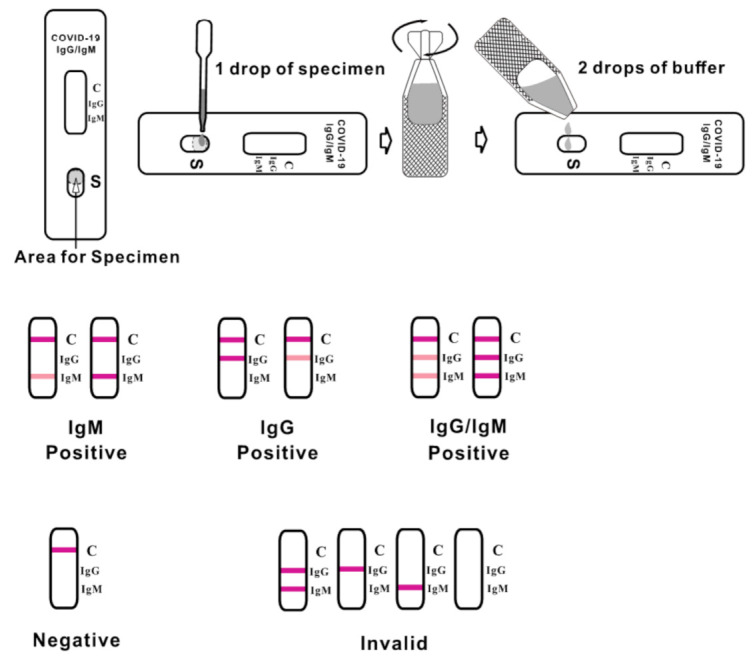
CLUNGENE^®^ Point-of-Care test manufacturer’s instructions.

**Table 1 healthcare-09-01124-t001:** Common infectious agents used in cross-reactivity study.

Analog
Anti-HBe, anti-HBc and HBsAg
Anti-Hepatitis C Virus (HCV)
Anti-Human Immunodeficiency Virus (HIV)-1
Anti-HIV-2
Anti-influenza A IgG
Anti-influenza B IgG
Anti-influenza A IgM
Anti-influenza B IgM
Anti-respiratory syncytial virus (RSV) IgG
Anti-respiratory syncytial virus (RSV) IgM
Anti-Mycoplasma pneumoniae (MP) IgM
Anti-Chlamydia pneumoniae (CP) IgM
Human parainfluenza virus PCR positive (Paired Convalescent)
Anti-Treponema pallidum (TP)
Rheumatoid factor (RF) 521.00 IU/mL RF 342.50 IU/mL RF 347.00 IU/mL RF 310.00 IU/mL RF 565.00 IU/mL RF 125.00IU/mL RF 796.00 IU/mL RF 500.00 IU/mL RF 825.00 IU/mL RF 144.00 IU/mL RF 158.00 IU/mL RF 122.00 IU/mL RF 197.00 IU/mL RF 146.00 IU/mL
Antinuclear antibodies (ANA) 1:240
Common human pathogenic coronaviruses: HCoV-HKU1 HCoV-NL63 HCoV-OC43 HCoV-229E

**Table 2 healthcare-09-01124-t002:** Reagents: reference numbers C58961 and C58957 Beckman Coulter Immunoassay System.

Reagent	Reference Number	Material Type
Access SARS-CoV-2 IgG Calibrator	Ref. No. C58963	
Access SARS-CoV-2 IgG Calibrator	Ref. No. C58958	
Access SARS-CoV-2 IgG QC	Ref. No. C58964	Quality Control (QC)
Access SARS-CoV-2 IgM QC	Ref. No. C58959	Quality Control (QC)
Access Substrate	Ref. No. 81906	Quality Control (QC)
Access Wash Buffer II	Ref. No. A16792	Quality Control (QC)
UniCel DxI Wash Buffer II	Ref. No. A16793	Quality Control (QC)

**Table 3 healthcare-09-01124-t003:** Quality control for DXI600 analysis of cross-reactivity.

Rack/Pos.	Sample ID Patient/Lot ID	Type Dilution	Test	Calibrator Results ^1^		RLUs ^2^	Completion
207/1	COV-2IgG QC1 922605	Serum	COV2G	Non-Reactive	0.02 S/CO	7143	27 April 2021 10:20 AM
207/2	COV-2IgG QC2 922605B	Serum	COV2G	Reactive	2.49 S/CO	995255	27 April 2021 10:20 AM
207/3	COV-2IgM QC1 922821	Serum	COV2M	Non-Reactive	0.16 S/CO	11220	27 April 2021 10:27 AM
			COV2M(2)	Non-Reactive	0.16 S/CO	11053	27 April 2021 10:27 AM
207/4	COV-2IgM QC2 922821	Serum	COV2M	Reactive	4.23 S/CO	301471	27 April 2021 10:28 AM
			COV2M(2)	Reactive	2.62 S/CO	186854	27 April 2021 10:27 AM

^1^ S/CO: Signal/Cutoff. ^2^ RLUs: Relative Light Units.

**Table 4 healthcare-09-01124-t004:** Antibody percent agreement.

Group	Days Post Symptom Onset	#PCR Total	Candidate Device Results		
			Results	PPA	95% CI
IgG Positive	0~7	4	2	50.00%	15.00~85.00%
	8~14	23	23	100.00%	85.69~100.00%
	≥15	35	33	94.28%	81.39~98.42%
	Total	62	58	93.55%	84.55~97.46%
IgM Positive	0~7	4	3	75.00%	30.06~95.44%
	8~14	23	21	91.30%	73.21~97.58%
	≥15	35	27	77.14%	60.98~87.93%
	Total	62	51	82.26%	70.96~89.79%
IgG/IgM Combined Antibody Positive	0~7	4	3	75.00%	30.06~95.44%
	8~14	23	23	100.00%	85.69~100.00%
	≥15	35	34	97.14%	85.47~99.49%
	Total	62	60	96.77%	88.98~99.11%
IgG Negative	N/A	37	37	100.00%	90.60~100.00%

**Table 5 healthcare-09-01124-t005:** Line Table Data.

# and Site	Subject ID	Age	Gender (F/M)	Whole Blood Specimen Collection Date	Days after Symptom Onset	CLUNGENE^®^ Rapid Test Result (Pos/Neg)		PCR Test Date	PCR Confirmation Result
						IgM	IgG		
1a	007sgh	66	F	1 June 2020		Neg	Neg	30 May 2020	Neg
2a	008sgh	65	F	1 June 2020		Neg	Neg	29 May 2020	Neg
3a	009sgh	72	M	1 June 2020		Neg	Neg	31 May 2020	Neg
4a	010sgh	67	F	5 June 2020		Neg	Neg	27 May 2020	Neg
5a	011sgh	44	M	5 June 2020		Neg	Neg	1 June 2020	Neg
6a	012sgh	31	F	5 June 2020		Neg	Neg	1 June 2020	Neg
7a	013sgh	77	F	5 June 2020		Neg	Neg	1 June 2020	Neg
8a	024smh	37	M	5 June 2020	11	Neg	Pos	2 June 2020	Pos
9a	025sgh	89	M	5 June 2020		Neg	Neg	11 June 2020	Neg
10a	023sgh	91	F	6 June 2020		Neg	Neg	9 June 2020	Neg
11a	024sgh	81	M	8 June 2020		Neg	Neg	9 June 2020	Neg
12a	026sgh	59	M	8 June 2020		Neg	Neg	11 June 2020	Neg
13a	019sgh	69	F	9 June 2020	20	Neg	Pos	23 May 2020	Pos
14a	028sgh	58	M	10 June 2020		Neg	Neg	11 June 2020	Neg
15a	021sgh	63	F	11 June 2020	13	Neg	Pos	6 June 2020	Pos
16a	027sgh	77	F	11 June 2020		Neg	Neg	1 June 2020	Neg
17a	029sgh	29	M	12 June 2020	8	Pos	Pos	8 June 2020	Pos
18a	033sgh	69	M	12 June 2020		Neg	Neg	5 June 2020	Neg
19a	034sgh	53	M	12 June 2020		Neg	Neg	11 June 2020	Neg
20a	035sgh	77	M	12 June 2020		Neg	Neg	11 June 2020	Neg
21a	036sgh	74	M	12 June 2020		Neg	Neg	10 June 2020	Neg
22a	037sgh	30	F	12 June 2020		Neg	Neg	9 June 2020	Neg
23a	038sgh	22	M	12 June 2020		Neg	Neg	6 June 2020	Neg
24a	001sgh	38	M	15 June 2020	26	Pos	Pos	23 May 2020	Pos
25a	006sgh	53	M	18 June 2020	23	Neg	Pos	29 May 2020	Pos
26a	027smh	47	M	18 June 2020	14	Pos	Pos	14 June 2020	Pos
27a	039sgh	60	M	18 June 2020	13	Pos	Pos	14 June 2020	Pos
28a	043sgh	22	F	18 June 2020	18	Neg	Pos	14 June 2020	Pos
29a	040sgh	70	M	19 June 2020	17	Pos	Pos	12 June 2020	Pos
30a	044sgh	32	F	19 June 2020		Neg	Neg	17 June 2020	Neg
31a	045sgh	66	M	19 June 2020		Neg	Neg	10 June 2020	Neg
32a	046sgh	65	M	19 June 2020		Neg	Neg	9 June 2020	Neg
33a	050sgh	31	M	26 June 2020	8	Pos	Pos	18 June 2020	Pos
34a	051sgh	44	M	26 June 2020	10	Pos	Pos	21 June 2020	Pos
35a	052sgh	35	M	26 June 2020	7	Pos	Pos	19 June 2020	Pos
36a	055sgh	57	M	26 June 2020		Neg	Neg	21 June 2020	Neg
37a	056sgh	69	F	26 June 2020		Neg	Neg	22 June 2020	Neg
38a	057sgh	67	M	26 June 2020		Neg	Neg	21 June 2020	Neg
39a	058sgh	59	F	26 June 2020		Neg	Neg	17 June 2020	Neg
40a	059sgh	39	M	26 June 2020	6	Neg	Neg	20 June 2020	Pos
41a	060sgh	28	F	26 June 2020		Neg	Neg	24 June 2020	Neg
42a	061sgh	51	F	26 June 2020		Neg	Neg	23 June 2020	Neg
43a	062sgh	71	F	26 June 2020	11	Pos	Pos	23 June 2020	Pos
44a	072sgh	62	M	2 July 2020	17	Pos	Pos	15 June 2020	Pos
45a	031smh	31	F	2 July 2020	22	Pos	Pos	11 June 2020	Pos
46a	066sgh	43	F	2 July 2020	15	Pos	Neg	24 June 2020	Pos
47a	068sgh	60	M	2 July 2020	12	Pos	Pos	25 June 2020	Pos
48a	075sgh	62	F	2 July 2020	7	Pos	Pos	25 June 2020	Pos
49a	081sgh	73	M	9 July 2020	9	Pos	Pos	4 July 2020	Pos
50a	082sgh	63	M	9 July 2020	11	Pos	Pos	3 July 2020	Pos
51a	084sgh	68	M	9 July 2020	15	Pos	Pos	2 July 2020	Pos
52a	088sgh	56	F	9 July 2020	10	Pos	Pos	7 July 2020	Pos
53a	090sgh	88	F	9 July 2020	20	Neg	Pos	7 July 2020	Pos
54a	089sgh	56	F	17 July 2020	16	Pos	Pos	4 July 2020	Pos
55a	094sgh	19	F	17 July 2020	15	Pos	Pos	11 July 2020	Pos
56a	096sgh	36	M	17 July 2020	27	Pos	Pos	14 July 2020	Pos
57a	097sgh	unknown	F	17 July 2020		Neg	Neg	14 July 2020	Neg
58a	098sgh	unknown	M	17 July 2020		Neg	Neg	8 July 2020	Neg
59a	099sgh	48	F	21 July 2020	9	Pos	Pos	18 July 2020	Pos
60a	100sgh	48	F	21 July 2020	14	Pos	Pos	17 July 2020	Pos
61a	101sgh	72	M	21 July 2020	16	Neg	Neg	25 July 2020	Pos
62a	053sgh	50	F	23 July 2020	27	Neg	Pos	26 June 2020	Pos
63a	073sgh	38	M	23 July 2020	34	Pos	Pos	30 June 2020	Pos
64a	103sgh	29	M	24 July 2020	12	Pos	Pos	15 July 2020	Pos
65a	104sgh	40	M	24 July 2020	21	Pos	Pos	11 July 2020	Pos
66a	106sgh	45	M	28 July 2020	15	Neg	Pos	22 July 2020	Pos
67a	113sgh	46	F	31 July 2020	7	Pos	Neg	25 July 2020	Pos
68a	114sgh	79	F	4 August 2020	12	Pos	Pos	30 July 2020	Pos
69a	110sgh	51	F	5 August 2020	15	Pos	Pos	29 July 2020	Pos
70a	109sgh	63	M	7 August 2020	19	Pos	Pos	25 July 2020	Pos
71a	119sgh	60	F	8 August 2020	10	Pos	Pos	7 August 2020	Pos
72a	105sgh	38	M	12 August 2020	31	Pos	Pos	20 July 2020	Pos
73a	120sgh	73	M	13 August 2020	22	Pos	Pos	26 July 2020	Pos
74a	123sgh	47	M	13 August 2020	12	Pos	Pos	10 August 2020	Pos
75a	124sgh	80	M	13 August 2020	30	Pos	Pos	20 July 2020	Pos
76a	118sgh	94	F	14 August 2020	15	Neg	Pos	31 July 2020	Pos
77a	117sgh	28	F	14 August 2020	14	Pos	Pos	5 August 2020	Pos
78a	121sgh	62	F	17 August 2020	14	Pos	Pos	12 August 2020	Pos
1b	C011	29	F	12 December 2020	14	Pos	Pos	30 November 2020	Pos
2b	A005	44	F	12 December 2020		Neg	Neg	6 December 2020	Neg
3b	C020	30	F	12 December 2020	14	Pos	Pos	28 November 2020	Pos
4b	C012	52	M	14 December 2020	14	Pos	Pos	5 December 2020	Pos
5b	C032	46	M	14 December 2020		Neg	Neg	10 December 2020	Neg
6b	C033	18	M	14 December 2020		Neg	Neg	10 December 2020	Neg
7b	C004	74	M	19 December 2020	26	Pos	Pos	3 December 2020	Pos
8b	C030	28	M	19 December 2020		Neg	Neg	16 December 2020	Neg
9b	C016	39	F	21 December 2020	19	Pos	Pos	10 December 2020	Pos
10b	C031	18	F	21 December 2020		Neg	Neg	16 December 2020	Neg
11b	C009	53	F	29 December 2020	18	Pos	Pos	19 December 2020	Pos
12b	C010	55	M	29 December 2020	19	Pos	Pos	19 December 2020	Pos
13b	C028	46	M	29 December 2020	24	Pos	Pos	18 December 2020	Pos
14b	C005	19	F	31 December 2020	21	Pos	Pos	18 December 2020	Pos
15b	C006	19	F	31 December 2020	21	Pos	Pos	18 December 2020	Pos
16b	C007	42	F	31 December 2020	21	Pos	Pos	18 December 2020	Pos
17b	C019	42	F	31 December 2020	15	Pos	Pos	18 December 2020	Pos
18b	C026	51	F	31 December 2020		Neg	Neg	31 December 2020	Neg
19b	C027	43	F	31 December 2020		Neg	Neg	31 December 2020	Neg
20b	C029	54	M	31 December 2020	15	Pos	Pos	26 December 2020	Pos
21b	C013	40	M	4 January 2021	28	Pos	Pos	15 December 2020	Pos

## Data Availability

The data presented in this study are available on request from the corresponding author. The data are not publicly available.

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
