# Peer review of "Updated Clinical Evaluation of the CLUNGENE® Rapid COVID-19 Antibody Test"

_healthcare, 2021, doi:10.3390/healthcare9091124_

Round 1
Reviewer 1 Report
I consider the manuscript worthy for publication, as it has some very current strengths in the dramatic scenario of the COVID-19 pandemic: in this regard, the spread of rapid kits for the diagnosis of positivity to SARS-CoV2 represents an undoubted advantage in terms commercial availability of diagnostic aids. A further strength is represented by the very simple method even for less experienced healthcare professionals.
Regarding weak points, I suggested:
- in the "Limitations of the study" section, it is necessary to specify that a further limitation is that the CLUNGENE antibody test has not been compared with another test (including commercial) for the rapid diagnosis of Covid-19, in terms of sensitivity and specificity. This aspect deserves to be clarified, since the difference in sensitivity and specificity of a test is a fundamental issue in terms of reliability, even for rapid tests;
- In materials and methods, for the medical use of the test in terms of reliability and storage, it would be important that the authors describe how they stored these tests, highlighting the storage temperature, if they are to be protected from direct light, etc. This information is potentially useful for large-scale use and in disadvantaged medical facilities.
Author Response
- Acknowledged the limitation and added to paper. See line 229. “Lastly, the CLUNGENE antibody test has not been compared with another test in the study.”
-
Added storage details. See line 82. “The CLUNGENE® test can be stored at ambient temperature between 4 and 30 Celsius. The product is packaged in a box with 25 test kits with each kit wrapped in an individual foil patch for proper storage and transportation. The test kit itself should be kept away from direct sunlight.”
Thank you in for the feedback!
Reviewer 2 Report
I found the paper interesting and helpful. It is generally understandable for a lay public based on the discussion section.
The only real question I had when reading was why in the Methods section 2.2.2 they mention the name of the person from whom they received samples rather than naming a laboratory or company. It seemed unusual. I did not see this as a major flaw. The discussion, including limitations was well written and identified usefulness and weakness of the antibody test.
Author Response
Clarified Trina Bio as a company. See line 100. “]; Trina Bioreactives, Ag, is an ISO certified company providing specialized Sars-COV-2 in vitro diagnostic biomaterials (https://trinabio.com/).”
Thank you for the feedback!